# DARTS: An Algorithm for Domain-Associated Retrotransposon Search in Genome Assemblies

**DOI:** 10.3390/genes13010009

**Published:** 2021-12-21

**Authors:** Mikhail Biryukov, Kirill Ustyantsev

**Affiliations:** Sector of Molecular and Genetic Mechanisms of Regeneration, Institute of Cytology and Genetics SB RAS, 630090 Novosibirsk, Russia; birykov@bionet.nsc.ru

**Keywords:** LTR retrotransposons, retroelements, domain annotation, software, automatic pipeline

## Abstract

Retrotransposons comprise a substantial fraction of eukaryotic genomes, reaching the highest proportions in plants. Therefore, identification and annotation of retrotransposons is an important task in studying the regulation and evolution of plant genomes. The majority of computational tools for mining transposable elements (TEs) are designed for subsequent genome repeat masking, often leaving aside the element lineage classification and its protein domain composition. Additionally, studies focused on the diversity and evolution of a particular group of retrotransposons often require substantial customization efforts from researchers to adapt existing software to their needs. Here, we developed a computational pipeline to mine sequences of protein-coding retrotransposons based on the sequences of their conserved protein domains—DARTS (Domain-Associated Retrotransposon Search). Using the most abundant group of TEs in plants—long terminal repeat (LTR) retrotransposons (LTR-RTs)—we show that DARTS has radically higher sensitivity for LTR-RT identification compared to the widely accepted tool LTRharvest. DARTS can be easily customized for specific user needs. As a result, DARTS returns a set of structurally annotated nucleotide and amino acid sequences which can be readily used in subsequent comparative and phylogenetic analyses. DARTS may facilitate researchers interested in the discovery and detailed analysis of the diversity and evolution of retrotransposons, LTR-RTs, and other protein-coding TEs.

## 1. Introduction

Transposable elements (TEs) are important players in the evolution of genomes [1,2,3,4]. The activity of TEs drives genetic diversity, contributes to the establishment of new gene regulatory networks and the rewiring of the existing ones, and can result in the origin of new genes sequestered by the host genome for its functioning [5,6,7]. The long-term existence and evolution of TEs has resulted in a broad diversity of the mechanisms for their transposition and replication, and the origin of a variety of different structural variants [8,9].

Retrotransposons, a group of TEs that move through a reverse transcription mechanism, are the most ubiquitous TEs in eukaryotic genomes. Due to their propensity to increase in copy number, retrotransposons constitute a substantial portion of the host genome, reaching as high as 80% of the total genome size in some plants [10,11]. Thus, studying retrotransposons is an essential part of understanding plant evolution. The majority of retrotransposons in plants are long terminal repeat (LTR) retrotransposons (LTR-RTs), which are structurally and evolutionarily similar to retroviruses of vertebrates [12,13]. Autonomous, i.e., capable of self-replication, LTR-RTs are complex genetic entities consisting of several protein-coding domains and non-coding regulatory sequences (such as LTRs) that mediate transcription, replication, and integration of the TEs [14,15,16,17]. Despite similarities in the general replication mechanism, LTR-RTs are structurally diverse and encode for additional protein domains, which are supposed to fine-tune their life cycle [18,19,20]. Despite structural differences, the central functional domain of all autonomous retrotransposons, reverse transcriptase (RT), remains conserved through evolution, allowing unbiased phylogenetic delineation and classification of retrotransposon diversity [9]. Evolution of LTR-RTs and other retrotransposons as individual entities attracts attention by itself, being an example of modular evolution [21,22]. In modular evolution, the main driving force is not a random mutational process, but acquisition, reshuffling, and loss of whole structural elements, such as protein domains and transcriptional enhancers. Therefore, the history of a distinct protein domain in a retrotransposon can be different from the evolution of its core RT domain [21,23,24].

The majority of computational tools developed for the annotation of LTR-RTs in genomic sequences initiate their search from identification of LTRs, and not conserved protein domains [25,26,27]. Alternative approaches, such as RepeatModeler, first look for any repetitive sequences on the nucleotide sequence level and then try to classify them based on the homology information to known TEs [28]. However, in cases when it is important to search for a specific family of TEs, these methods, apart from being too redundant and computationally time-consuming, may end up with a very high rate of false-negative results, since some TE lineages may be present in a very low number of copies, and some LTR-RT copies may lack well-detectable LTRs. On the other hand, homology-based approaches suffer from the incompleteness of the reference databases [29].

Here, based on our experience in retrotransposon identification [21,23], we developed a new computational tool that takes advantage of the conserved nature of protein domains encoded by retrotransposons—DARTS (an algorithm for Domain-Associated Retrotransposon Search in Genome Assemblies). DARTS uses an open and actively supported database of conserved protein domain sequence profiles instead of relying on databases of representative reference elements. By selecting a certain set of sequence profiles, DARTS can be easily customized for the identification of virtually any group of TEs with a known conserved protein domain sequence, a model of which is present in the database. Additionally, DARTS performs structural annotation and extraction of sequences of the corresponding protein domains. The extracted sequences can be readily used in subsequent comparative and phylogenetic analyses to study the evolution of a particular group of TEs in more detail.

## 2. Materials and Methods

### 2.1. Data Collection

For the analysis, we downloaded genome reference assemblies from the NCBI Genome database (https://www.ncbi.nlm.nih.gov/genome/, accessed on 23 November 2021) of four model plant species: *Arabidopsis thaliana* (TAIR10.1, 120 Mbp), *Nicotiana tabacum* (Ntab-TN90, 3736 Mbp), *Selaginella moellendorffii* (GCF_000143415.4, 212 Mbp), and *Zea mays* (Zm-B73-REFERENCE-NAM-5.0, 2192 Mbp).

### 2.2. Description of the DARTS Pipeline

The DARTS pipeline consists of several scripts written in Python (v 3.6) and Bash programming languages. The scripts, installation, and detailed usage manuals are available on GitHub: https://github.com/Mikkey-the-turtle/DARTS_v0.1, accessed on 23 November 2021. The general pipeline scheme is shown in Figure 1. A user may choose which parts of the pipeline to execute and can customize every filtering and threshold value presented in the default version, which was originally adapted for identification of LTR-RTs.

Before the analysis, DARTS checks the total genome assembly size, and, if it exceeds 1 Gbp, splits the assembly into several smaller batches to allow for a fast search. If the split is required, the program will attempt to divide the file into individual chromosomes (scaffolds or contigs) without disrupting the original sequences. However, if the genome assembly contains long sequences, such as fully-assembled chromosomes that exceed 1 Gbp in size, the sequences are divided into batches below 100 Mbp creating no more than N-1 breakpoints, where N equals to the number of batches formed from the chromosome. At the same time, chromosomes below 1 Gbp are left intact and put into separate corresponding batches.

To identify target protein domains, DARTS uses standalone Reverse PSI-BLAST (RPS-BLAST) from the BLAST+ package [30] supplemented with corresponding multiple sequence alignment protein models, or profiles, obtained from a local copy of the NCBI Conserved Domain Database (CDD) [31] (https://www.ncbi.nlm.nih.gov/Structure/cdd/cdd.shtml, accessed on 23 November 2021). Two sets of CDD profiles related to a certain group of TEs must be defined by the user prior to the analysis. The first set is a single CDD profile representing a core domain of the TE group (e.g., reverse transcriptase domain specific to LTR-RTs). The second set contains all other additional CDD profiles expected to be found in TEs of interest as well as the first CDD profile. In the first RPS-BLAST search round, the genomic assembly is scanned by the first CDD profile. This results in a set of matches that are pre-filtered by *e-value* (1 × 10^−3^) and length of the match. The genomic coordinates of a match are identified, and the corresponding nucleotide sequence with flanking regions of 7500 bp in length is extracted for each match. The second RPS-BLAST search utilizes the second user-defined set of CDD profiles and is applied on the extracted sequence regions instead of the whole assembly. Processing of the second RPS-BLAST run results in the structural annotation of TEs of interest and subsequent filtration of the elements by presence of a user-defined set of protein domains. Importantly, when several core domains are present in the same expanded matching region, DARTS will try to delineate them into separate domain assemblies. When a domain match is interrupted by frameshifts or small insertions, DARTS will assemble its parts in a single unit for annotation (Appendix A). Amino acid sequences of each of the identified domains are extracted and stored in separate FASTA-formatted files. For LTR-RTs, using the BLASTn tool from the BLAST+ package [30], DARTS will attempt to identify LTRs flanking the first and the last identified protein domains with more than 80% identity that are more than 100 bp but less than 3000 bp in length. Each element obtains a score (%score) based on the number of identified protein domains, length and quality of the matches, presence of uninterrupted open reading frames (ORFs), and LTR identity for LTR-RTs. Each sequence that passed the filtration stage will have a unique name identifier presented in the following format: “%project_name_%batch_%num_ID|%structure|%LTR_information|%score”, where %project_name is the user-defined name of the DARTS run, %batch is the number of the corresponding genome batch-file, %num_ID is the numerical identifier in the current genome batch-file, %structure is the generalized protein domain-based structure presented for Ty3/gypsy LTR-RTs (e.g., “GAG.Pro.gRT.gRH.INT”), %LTR_information is shown as LTR%identity-length (e.g., LTR%99.567-232), and %score is the float number.

For the purpose of subsequent comparative and phylogenetic analyses, DARTS can reduce the redundancy of the dataset through clustering using MMseqs2 [32] and subsequent selection of clusters’ representatives based on the %score value and structural composition. Clustering information is stored as a tab-separated values table file and can later be reanalyzed using custom criteria. For LTR-RTs, clustering is performed by default using the core and the most conserved domain, reverse transcriptase (RT), with the following default parameters: “easy-cluster -min-seq-id 0.8 -c 0.8”. Nucleotide sequences of the representative elements and amino acid sequences of each of their protein domains are deposited in separate FASTA-formatted files. These sequences can later be directly used for multiple sequence alignment generation for subsequent phylogenetic analysis.

### 2.3. Identification of LTR Retrotransposons Using LTRharvest

To mine LTR-RTs from the selected plant genomes using the de novo LTR-RT prediction tool LTRharvest [26], we ran the program with the following parameters: “-minlenltr 200, -maxlenltr 2000, -mindistltr 3000, -maxdistltr 22000, -similar 85.0, -overlaps no, -mintsd 3, -maxtsd 20”. The resulting file with all the hypothetical full-length LTR-RT nucleotide sequences produced by LTRharvest was then processed by DARTS to identify sequences containing the RT domain and to ensure unbiased comparison between both tools. To compare the number of elements uniquely identified by both the DARTS and LTRharvest tools, we performed reciprocal BLASTn searches with a “-max_target_seqs 1” parameter.

## 3. Results

Previously, we performed a study on the diversity and evolution of a structurally variable group of Ty3/gypsy plant LTR-RTs—Tat [23,33,34]. Tat LTR-RTs have an additional ribonuclease H domain (aRNH) of the so-called archaeal origin, which is fixed in several positions with regard to other domains in different Tat lineages [23]. In our previous study on Tat [23], we used a conventional tool for the de novo prediction of LTR-RTs—LTRharvest [26]. Later, when doing an independent search using tBLASTn with an aRNH sequence as a query, we found that a substantial fraction of aRNH-containing Tat LTR-RTs were underrepresented in the LTRharvest output. We reasoned that this could be explained by the majority of LTR-RT copies in the studied plant genomes being damaged, fragmented (not intact), and lacking detectable LTRs. The fact that LTRs are used as a starting point for LTR-RT identification in several published software [25,26,27,35], including LTRharvest (now a part of the most popular de novo repeat identification pipeline RepeatModeler [28]), inspired us to develop a new algorithm that could automatically perform identification of protein-coding TEs and LTR-RTs in particular. We named it Domain-Associated Retrotransposon Search (DARTS), as the initiation of the screen and subsequent structural annotation are based on the prediction of conserved protein domains and not LTR sequences. The basis for DARTS is our experience in semi-automated identification of both protein-coding LTR-RTs and non-LTR retrotransposons [21], as well as conceptually similar approaches performed by other researchers [20,36,37].

For the analysis, we selected reference genome assemblies of four widely used model plant species varying in genome size and TE content (see Section 2.1) and applied DARTS and LTRharvest to identify LTR-RTs. The DARTS search was initiated with the most conserved reverse transcriptase (RT) domain, while the LTRharvest attempts to identify regions flanked by direct repeat sequences of hypothetical LTRs [26]. Using DARTS, we mined 267,105 LTR-RT elements (88,389 with LTRs) in the four studied genomes, while only 34,030 sequences predicted by LTRharvest contained the RT domain sequence after filtration of the 55,658 elements originally predicted (Figure 2A). Importantly, all the 34,030 LTRharvest elements were also predicted by DARTS (Appendix A), suggesting almost eight times higher sensitivity of the latter.

To exemplify DARTS performance when search is initiated from a different protein domain, we screened for *Tat* LTR-RTs in the same genome assemblies using the aRNH CDD profile in the first round of RPS-BLAST. DARTS found 59,082 elements (20,171 with LTRs), while only 11,529 LTR-RTs were identified by LTRharvest (Figure 2B). This suggests that the overall abundance of *Tat* LTR-RTs in our previous study using LTRharvest [23] was largely underestimated.

It must be noted that potential false-positive matches can be present when only the initial target domain is found. However, the chances of this are low since, during the second step of the RPS-BLAST search, all the domains are re-annotated again, which results in an increase of *e-value* since the size of the database is decreased to a single sequence region. Nevertheless, the false-positive hits can be filtered out on the way to phylogenetic analysis, standing as outliers during clustering and multiple sequence alignment compared to true-positive representatives. Alternatively, whenever it is possible, we would suggest filtering the results of DARTS by the presence of one or two additional domains or regulatory sequences, such as LTRs, to completely avoid the problem. In this study, we found that the number of RT-only containing matches in the RT domain search initiated by DARTS equaled 5.8% ± 2.5% (mean ± standard deviation of the mean). Therefore, this range can be considered as a theoretical stringent upper boundary for the false-positive TEs detected by DARTS.

## 4. Discussion

Although we have shown examples of DARTS usage for general LTR-RT identification and targeted Tat LTR-RT mining in plants, our primary object of interest, the software can be easily customized for search of other TEs with conserved protein-coding domains in other eukaryotic genomes. For example, *Penelope*-like retroelements can be targeted by search for their specific RT and endonuclease domains [38,39] and DIRS-like retrotransposons by their RT and tyrosine recombinase domains [40,41]. Apart from their specific RT domain, various non-LTR retrotransposon groups have two types of endonucleases and two types of RNH domains [23,37,42]. Cut-and-paste DNA transposons can be identified by the transposase domains, e.g., DNA helicases can be found in *Helitrons* and DNA polymerases in *Mavericks* [43,44,45].

For non-LTR retrotransposons and DNA transposons, the DARTS initial identification approach is similar to the methods implemented in previously published software, such as MGEScan-non-LTR and TransposonPSI [36,46]. However, DARTS is more advantageous since it can also perform automatic structural annotation and clustering, and its algorithm relies on the actively supported RPS-BLAST tool and the CDD database. Therefore, more sensitive profiles can be used to provide a detailed and targeted annotation of TEs of interest. Additionally, compared to TransposonPSI, DARTS returns amino acid sequences of each of the identified domains without the need for additional parsing, allowing direct transition to phylogenetic analysis.

A part of TE analysis that is not covered by DARTS is the annotation of non-protein coding genes and copies lacking the domain of interest that was used to initiate the search. While annotation of such elements as Short Interspersed Nuclear Elements (SINEs) and Miniature Inverted-repeat Transposable Elements (MITEs) indeed requires a substantially different approach for their identification [28,47], severely damaged copies and sequences, such as solo-LTRs, lacking a domain of interest can still be found by applying nucleotide BLAST or RepeatMasker [30,48] using the copies identified by DARTS as queries. Thus, their number can be accounted for in the genome annotation.

## 5. Conclusions

Here, we developed a new pipeline for automatic search and structural annotation of protein-coding LTR-RTs and other retrotransposons in genomic sequences. DARTS is beneficial when one is interested in analysis of all the structural diversity of a TE group. We showed that DARTS is almost eight times more sensitive in LTR-RT identification than the de novo tool LTRharvest, which is now included in the widely used RepeatModeler version 2 pipeline [28]. The ease of DARTS customization should facilitate many researchers studying the diversity and evolution of different groups of TEs.

## Figures and Tables

**Figure 1 genes-13-00009-f001:**
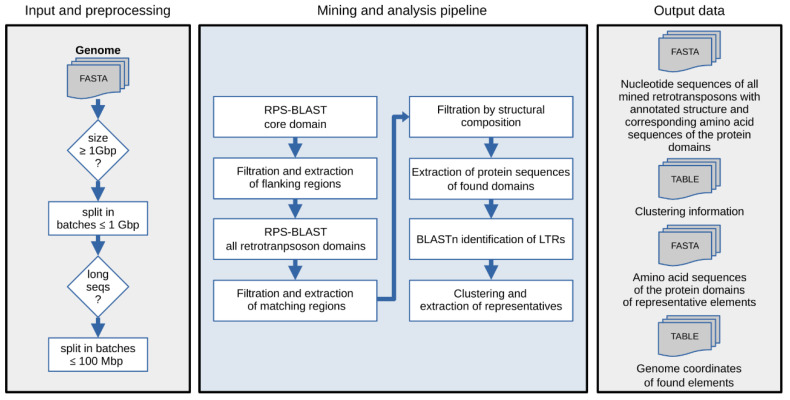
Principle scheme of the DARTS (Domain-Associated Retrotransposon Search) workflow. Detailed description of each step is in the text.

**Figure 2 genes-13-00009-f002:**
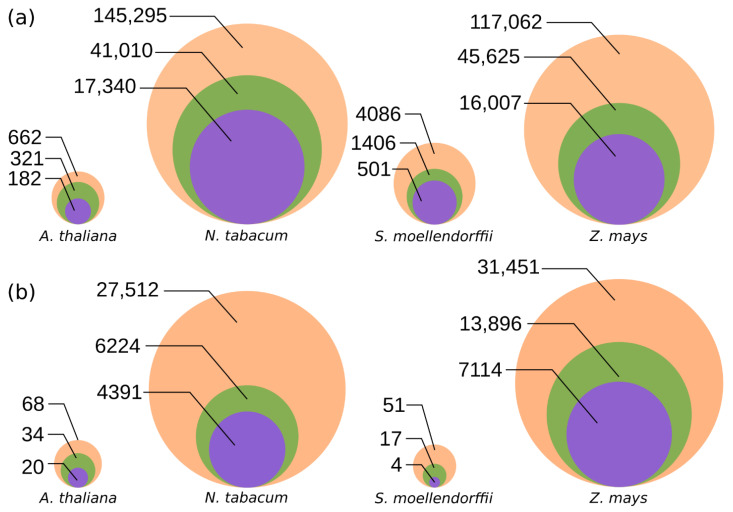
Sensitivity of LTR retrotransposon (LTR-RT) identification by the DARTS and LTRharvest pipelines. Orange circles—number of elements found by DARTS; green circles—number of LTR-RTs found by DARTS with predicted LTRs; purple circles—number of elements found by LTRharvest. Sizes of the circles are proportional to the number of elements with relation to the minimum and the maximum values. Exact numbers of elements are indicated to the left of the circles. Parameters of the LTRharvest search were the same for both the approaches (**a**,**b**). (**a**) Prediction of LTR-RTs by DARTS when the search was initiated from the RT domain; LTRharvest elements were retained if the RT domain was present. (**b**) Prediction of LTR-RTs by DARTS when the search was initiated from the aRNH domain; DARTS and LTRharvest elements were retained if both the RT and aRNH domains were present.

## Data Availability

The software and datasets produced in this study are openly available on GitHub https://github.com/Mikkey-the-turtle/DARTS_v0.1.

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
