# Peer review of "DARTS: An Algorithm for Domain-Associated Retrotransposon Search in Genome Assemblies"

_genes, 2021, doi:10.3390/genes13010009_

Round 1

Reviewer 1 Report

The manuscript presents an intriguing algorithm via DARTS (Domain Associated Retro Transposon Search in Genome Assemblies). I had a great time analyzing this study. I have a few questions for the authors to answer and see how it helps them present a better read of the manuscript.

Lines 66-68: How do the authors account for this when they say the database is constantly being updated and the reference databases are incomplete?

Line 69: Says any protein-coding group of TEs, which I assume means proteins whose structure and functional data are available in databases, rather than unidentified proteins? I would revise the sentence more carefully.

Lines 69-72: Again, this is a little perplexing. I might also paraphrase the sentence. And, most importantly, can we say that these are the features that distinguish DARTS, namely homology modeling of sequences for phylogeny analysis?

Lines 89 to 91: I'm curious how the parameters for splitting are defined. I'm concerned about how this might affect the sequence annotation process downstream.

Line 105: At what point in the process do we define this "user-defined set of protein domains"?

Lines 153-155: I'm curious if we can attribute damaged and fragmented genomes to the initial process of genome splitting. So, how do we address this with DARTS?

Line 166: Would it be interesting to discuss whether genome size and TE content matter when choosing the best tool for analysis?

Line 192: I'm afraid I've been misunderstood here. However, I believe the method section stated that the second step of the RPS-BLAST uses a database of all CDD profiles, including the first profile. So, how is the database's size reduced to a single sequence later?

Lines 218–219: Can we specify the type of database used by other existing software when we emphasize that DARTS works by constantly updating RPS-BLAST and the CDD database? Again, we'd be better off paraphrasing the phrase "constantly updating," which, unless otherwise stated, I believe every reliable database is.

Author Response

Dear Editor, dear Reviewers,

we were happy to receive your decision that only minor revisions are suggested. We would like to thank you for your incredible work for such a fast processing of our manuscript. We additionally want to thank the reviewers for their valuable comments and overall conclusion regarding the manuscript quality.

From the Editorial office we got one major comment where we were asked to split the “3. Results and Discussion” section into two separate sections: “3. Results” and “4. Discussion”, respectively. This is now done, and because of it we removed additional subsection headings “3.1. Application of DARTS for general and lineage-specific LTR retrotransposon identificationand “3.2. Possible application of DARTS for identification of other transposable elementsas redundant.

Apart from the edits which we made in response to one of the reviewer’s suggestions (see below), a few typos and style/grammar changes were fixed. All the edits were done using Track Changes in MS Word.

Below we present a point-by-point response to the Reviewer 1 comments.

Block 1: Introduction

1) “>>Lines 66-68: How do the authors account for this when they say the database is constantly being updated and the reference databases are incomplete?”

2) “>>Line 69: Says any protein-coding group of TEs, which I assume means proteins whose structure and functional data are available in databases, rather than unidentified proteins? I would revise the sentence more carefully.”

3) “>>Lines 69-72: Again, this is a little perplexing. I might also paraphrase the sentence. And, most importantly, can we say that these are the features that distinguish DARTS, namely homology modeling of sequences for phylogeny analysis?”

We thank the reviewer for pinpointing bad phrasing in this paragraph. We agree that it was perplexing and could have confused potential readers of the manuscript. DARTS is not a phylogenetic tool but it is ment to identify and extract sequences that later can be directly used for phylogenetic analysis and comparative analysis of structural composition of TEs. Therefore we rewrote the paragraph as follows:

DARTS uses an open and actively supported database of conserved protein domain sequence profiles instead of relying on databases of representative reference elements. By selecting a certain set of the sequence profiles, DARTS can be easily customized for identification of virtually any group of TEs with a known conserved protein domain sequence, a model of which is present in the database. Additionally, DARTS performs structural annotation and extraction of sequences of the corresponding protein domains. The extracted sequences can be readily used in subsequent comparative and phylogenetic analyses to study evolution of a particular group of TEs in more detail.

Block 2: Methods

4) “>>Lines 89 to 91: I'm curious how the parameters for splitting are defined. I'm concerned about how this might affect the sequence annotation process downstream.”

We agree with the reviewer that we did not describe the split procedure in enough detail. To avoid potential confusion of new readers we added the following information and edited the paragraph as follows:

Before the analysis, DARTS checks the total genome assembly size, and, if it exceeds 1 Gbp, splits the assembly into several smaller batches to allow fast search. If the split is required, the program will attempt to divide the file into individual chromosomes (scaffolds or contigs) without disrupting the original sequences. However, if the genome assembly contains long sequences such as fully-assembled chromosomes the size of which exceeds 1 Gbp, the sequences are divided into batches below 100 Mbp creating no more than N-1 breakpoints, where N equals to the number of batches formed from the chromosome. At the same time, chromosomes below 1 Gbp are left intact and put into separate corresponding batches.

5) “>>Line 105: At what point in the process do we define this "user-defined set of protein domains"?”

We agree with the reviewer that the way it was written makes it unclear. Therefore, we rewrote the paragraph as follow:

Two sets of CDD profiles related to a certain group of TEs must be defined by the user prior to the analysis. The first set is a single CDD profile representing a core domain of the TEs group (e.g. reverse transcriptase domain specific to LTR-RTs). The second set contains all other additional CDD profiles expected to be found in TEs of interest as well as the first CDD profile. In the first RPS-BLAST search round the genomic assembly is scanned by the first CDD profile. This results in a set of matches which are pre-filtered by e-value (1E-3) and length of the match. Genomic coordinates of a match are identified, and the corresponding nucleotide sequence with flanking regions of length 7500 bp is extracted for each match. The second RPS-BLAST search utilizes the second user-defined set of CDD profiles and is applied on the extracted sequence regions instead of the whole assembly.

Block 3: Results

6) “>>Lines 153-155: I'm curious if we can attribute damaged and fragmented genomes to the initial process of genome splitting. So, how do we address this with DARTS?”

We think that it should be more clear now how the splitting is performed (see the response to the point (4)). There is a very low chance that an element would occur at the point of split, and splits are generally rare and do not exceed N-1 where N is the number of batches derived after split. LTRharvest, however, in its pipeline does not perform any splitting and still returns a significantly lower number of LTR-RTs.

7) “>>Line 166: Would it be interesting to discuss whether genome size and TE content matter when choosing the best tool for analysis?”

We find this problem out of scope of this manuscript. Here, we are not aiming to find the best tool for analysis which would require thorough benchmarking of many tools and would always depend on a particular goal of the researcher. The aim of DARTS is to automate the procedure of identification and extraction of TEs with known conserved protein domains such as Tat LTR-RTs with the aRNH domain which were poorly identified by LTRharvest in our previous studies. As for the genome size, it really matters when one has a limited computational power. When we were testing DARTS (unpublished data), we found that thanks to splitting it can process in a few days such a gigantic genome as one of Ginkgo biloba (10,6 Gbs) (https://gigascience.biomedcentral.com/articles/10.1186/s13742-016-0154-1). This is a task that was just impossible for LTRharvest to complete even in a week time. However, we still think that to properly address this point a more thorough benchmarking should be done.

8) “>>Line 192: I'm afraid I've been misunderstood here. However, I believe the method section stated that the second step of the RPS-BLAST uses a database of all CDD profiles, including the first profile. So, how is the database's size reduced to a single sequence later?”

We think that it should be more clear now how the CDD profiles are selected (see the response to the point (5)). The database of CDD profiles is increased, indeed, but now the search is performed against single extracted sequence regions (target database size equals N bp of a single sequence) and not against the whole genome (target database size equals N bp of the whole genome). This affects E-value as by chance it is more probable to find any similarity in a larger pool of sequences than in a smaller one. For each CDD profile E-value for a hit is calculated independently, so the size of the query database does not matter here.

Block 4: Discussion

9) “>>Lines 218–219: Can we specify the type of database used by other existing software when we emphasize that DARTS works by constantly updating RPS-BLAST and the CDD database?

Again, we'd be better off paraphrasing the phrase "constantly updating," which, unless otherwise stated, I believe every reliable database is”

We agree with the reviewer that the phrasing in this sentence is poor. We rephrased it as follows, avoiding the misleading words “constantly updating”:

However, DARTS is more advantageous since it can also perform automatic structural annotation and clustering, and its algorithm relies on the actively supported RPS-BLAST tool and the CDD database.

---------------------

best regards,

Kirill Ustyantsev

Reviewer 2 Report

The manuscript presents data on the development of a computational pipeline to mine sequences of protein-coding retrotransposons based on the sequences of their conserved protein domains - DARTS. The authors achieved results that can be used not only in the presented manuscript, but also in the subsequent and in-detail analysis of diversity and evolution of retrotransposons, LTR-RTs, and other protein-coding TEs. The rationale for the establishment of this work ascertains its novelty, the extensiveness of the research, and its importance in the context of comparative and phylogenetic analyses. 

Author Response

(The authors gave the same response as above.)
